# Dispositional goal orientation and perceptions of coach motivational climate on attitudes towards doping among Kenyan endurance runners

Kevin Kiprotich Kipchumba[1]*, Elijah Rintaugu[2], Francis Mwangi[1], Benson Gathoni[1]

1 Department of Physical Education, Exercise and Sports Science, Kenyatta University, Nairobi, Kenya,
2 Department of Recreation and Sports Management, Kenyatta University, Nairobi, Kenya

* kevinkipchumba@gmail.com

## Abstract

Changing athletes' attitudes towards doping has been shown as crucial in prevention efforts in combating doping in sports, with dispositional goal orientation and perceptions of coach motivational climate identified as factors shaping doping attitudes among athletes. The purpose of this study was to examine the relationships between dispositional goal orientation and motivational climate on attitudes towards doping among Kenyan Endurance runners. A cross-sectional survey design was used to collect data from 323 Kenyan runners with 215 males (66.6%) and 108 females (33.3%). The study assessed athletes' goal orientation through the Task and Ego Goal Orientation Sport Questionnaire, perceptions of coach motivational climate through Perceived Motivational Climate in Sport Questionnaire, and attitudes towards doping through Performance Enhancement Attitude Scale. Descriptive statistics, correlation analysis, Mann-Whitney U tests and Kruskal-Wallis H tests was used for data analysis. The study found significant inverse relationships between mastery climate and doping attitudes (rho = -.242; $p$ <.001), as well as between task orientation and doping attitudes (rho = -.158; p <.004). Conversely, performance climate (rho =.362; $p$ <.001) and ego orientation (rho =.362; $p$ <.001) showed significant positive relationships with doping attitudes. There were no significant differences in doping attitudes based on age (U = 11582.500, $p$ <.191), gender (U = 11437.500, $p$ <.827) and athlete's length of experience ($\chi$2 (2) = 1.359, $p$ <.507). The study concludes that fostering mastery-oriented coach motivational climate and promoting task-oriented goal orientation could effectively cultivate anti-doping attitudes among athletes and enhance clean sport.

## 1. Introduction

Doping is a worldwide problem that affects both competitive and non-competitive athletes, posing serious negative consequences to athletes' health, as well as

**Data availability statement:** All relevant data are within the paper and its Supporting Information files.

**Funding:** The author(s) received no specific funding for this work.

**Competing interests:** The authors have declared that no competing interests exist.

threatening the integrity and reputation of sports [1,2]. Although, various preventive and deterrence measures have been instituted and implemented to combat this problem, changing athletes' attitudes towards doping is seen as crucial in prevention efforts [3,4]. Alongside these measures, studies have opined that identifying individual athletes and understanding situational factors that may positively or negatively influence athletes' doping attitudes may assist in doping prevention [5,6].

Attitude encompass dispositions and evaluations individuals possess regarding an object or thought [7]. Therefore, attitudes towards doping represents athletes' beliefs, values, and opinions regarding the utilization of prohibited substances and techniques to enhance performance in sports [8]. Attitudes ranges from strong anti-doping stances to more favourable views on doping. Research on athletes' attitudes towards doping indicates that athletes with positive attitudes toward doping are more inclined to engage in doping compared to those with negative attitudes [9,10]. Hence, understanding the factors that shape athletes' attitudes towards doping is crucial in formulation of effective interventions and prevention strategies towards preventing doping and protecting clean athletes [10].

Dispositional goal orientation and perceptions of coach motivational climate in sport are crucial constructs influencing athletes' attitudes towards doping. Achievement Goal Theory (AGT) [11], suggests that how individuals define success and competence (goal orientations) and how their social context is shaped (motivational climate) influence motivated behaviors in sports. This highlights the importance of understanding the interaction between personal goals and situational factors in predicting athletes' attitudes towards doping. Achievement Goal Theory (AGT) proposes two types of goal orientation: task and ego orientation [11]. Task orientation refers to individuals who focus on self-referential success, personal improvement, and effort and are more likely to adhere to rules, support fair play, and have negative attitudes towards doping, while ego-oriented individuals focus on outperforming others, winning at all costs, and demonstrating superior performance and exhibit more positive attitudes towards doping [10,12,13].

Motivational climate refers to the situational goal structure affecting athletes' task and ego involvement in sports ([14]. Coaches have been shown to play a significant role in creating the motivational climate for athletes [10,15,16]. For instance, coach created mastery climate emphasizes on cooperation, personal improvement, and effort, while coach created performance climate emphasizes on competition, intra-team rivalry, punishment for mistakes, unequal recognition of athletes, and athletes winning at all costs [15]. In regards to doping, empirical evidence has shown that coach created motivational climate that empowers athletes and a mastery climate created by coaches can lead to prosocial behavior, respect for the game, and anti-doping attitudes, while a performance climate and coach controlling style and disempowering climate can lead to anti-social behavior and pro-doping attitudes among athletes [10,16,17].

Research has shown inconsistent findings in the relationship between goal orientation, motivational climate and doping attitudes. For instance, athletes with task goal orientation and mastery motivational climate have been associated with negative attitudes towards doping and protective against doping behaviors, while athletes with ego goal orientation and performance motivational climate have been linked with

positive attitudes towards doping and risky attitudes towards drug use [10,18]. Similarly, task orientation has been reported to be weak and negative predictor of doping attitudes among athletes, while ego orientation is not related with doping attitudes [19]. Another inconsistency indicate that task orientation is not related with doping attitudes, while ego orientation is positively associated with doping attitudes [20]. Given demonstrated associations and inconsistent findings between goal orientations, motivational climate, and doping attitudes, little is known about these relationships among Kenyan endurance runners, particularly regarding how athletes assess their sporting success and perceive their coaching environment.

In terms of demographic factors, research has found that male athletes have positive doping attitudes, while female athletes possess negative doping attitudes. Female athletes cite reasons such as the belief that doping is wrong, fear of being banned from sport, concerns about fertility issues, and potential media exposure [21–23]. Conversely, male athletes are motivated by the desire for increased muscle mass and long-term financial stability [22]. However, studies in East Africa indicate that gender does not significantly influence doping intentions and attitudes among students and athletes [24–26]. In regards to age, older athletes tend to have more negative attitudes towards doping compared to younger athletes due to their greater knowledge and ability to make informed decisions about doping [26,27]. In contrast, younger student athletes have been reported to have negative perceptions about doping compared to other age groups [24]. Athletes' with more length of sporting experience have been reported to have negative attitudes towards doping as they are believed to have better understanding about doping compared to athletes with short length of sporting experience [27,28]. However, playing experience did not predict attitudes towards doping among East African athletes [26].

Kenya is a global leader in distance running but faces a growing doping issue, as evidenced by a high number of positive tests and sanctions against Kenyan athletes [29–31]. Further information from ADAK reveal that between August 5th, 2017, and September 14th, 2023, a total of 44 Kenyan athletes from different sport disciplines have faced sanctions ranging from bans of one year to lifetime from either ADAK or International Sports Federations due to violations of anti-doping rules [32]. Similar reports indicate that endurance runners in Kenya account for a majority of the positive doping tests compared to other sports disciplines [33]. This threatens Kenya's reputation as a sporting power and athletics which is regarded as Kenya's national heritage and the country's number one sport [31]. Research has cited various factors that contribute to doping among Kenyan athletes, including inadequate training on doping rules, the influence of athlete support personnel (ASP), and financial pressures such as sponsorships and endorsements [34,35]. While existing research has explored several factors influencing doping behaviors in athletes, there is a clear gap in understanding how dispositional goal orientation and coach created motivational climate influence doping attitudes, particularly among Kenyan endurance runners. This study sought to fill this gap by examining these factors within the Kenyan athletic context.

## 2. Materials and methods

### 2.1. Study design and Participants

The study targeted 2,000 Kenyan endurance runners competing in events from 800 meters to marathons including track, road, and cross-country races. A cross-sectional analytical survey design was used, with stratified random sampling ensuring proportional representation of participants by gender, age, and running experience, thereby enhancing the precision and generalizability of the findings. A total of 323 Kenyan endurance runners with 215 males (66.6%) and 108 females (33.3%) participated in the study. The sample size was determined using the [36] formula and WADA Social Science Research Package for Anti-Doping organizations supported this sample size [37]. Eligible participants included runners who had been training for at least 6 months, were at least 14 years old, and had no history of history of doping violations.

### 2.2. Measures

#### 2.2.1 Dispositional goal orientation. Dispositional goal orientation was assessed using the 13-item Task and Ego Orientation in Sport Questionnaire (TEOSQ) [38].Participants rated their agreement with the statement "I feel really

successful in athletics when...” followed by 13 questions related to task orientation (e.g., “when I work really hard” and “when I learn a new skill and it makes me want to practice more”) and ego orientation (e.g., “when others mess up and I don’t” and “when I can do better than my friends “). Responses were rated on a 5-point Likert scale from 1 (strongly disagree) to 5 (strongly agree). The Mean scores for both task and ego orientation subscales were calculated for analysis with higher scores indicating more characteristic goal orientations. [39] found good reliability index for the task subscale (α = 0.78) and ego subscale (α = 0.91).

**2.2.2. Perceptions of coach motivational climate.** Athletes’ perceptions of coach-created motivational climate were measured by the Perceived Motivational Climate in Sport Questionnaire (PMCSQ-2) [15]. The questionnaire consisted of 17 mastery climate items that contained three dimensions of the existence of important roles (“each athlete contributes in some important way”), cooperative learning (“athletes’ help each other learn”), and effort/improvement (“athletes feel good when they try their best) and 16 performance climate with three dimensions that that assessed unequal recognition (“only the best athletes get praise”), intra-team rivalry (“athletes are encouraged to outperform the other athletes”) and punishment for mistakes (“the coach gets mad when an athlete makes a mistake”). The 33 items were prefaced with the heading “In this athletics team**…**”. A five-point Likert-type scale, ranging from strongly disagree (1) to strongly agree (5), was used to answer the items. The Mean scores for both mastery climate and performance climate subscales were calculated for analysis with higher scores reflecting more characteristic motivational climates. [15] have shown good reliability index of the motivational climate (α = 0.87) and performance climate subscales (α = 0.89).

**2.2.3. Attitudes towards doping.** The 17-item Performance Enhancement Attitude Scale (PEAS) was used to measure athletes’ attitudes towards doping [8]. Participants used a Likert scale from 1 (strongly disagree) to 5 (strongly agree) to express their level of agreement with various statements. A higher score indicates a more positive attitude towards doping. Previous research has demonstrated strong reliability index of the scale, with reported Cronbach’s alpha values ranging from 0.71 to 0.91 [8,40].

## 2.3. Procedures

The study was conducted in accordance with the Declaration of Helsinki, and approved by the Kenyatta University Ethical Review Board (KUERB) under approval number PKU/1074/11124, issued on September 17, 2019 and research permit from National Commission for Science, Technology, and Innovation (NACOSTI) under license number NACOSTI/P/19/985. After receiving ethical approval from the Kenyatta University, athletes were recruited from various athletics clubs in Elgeyo-Marakwet County, Kenya. Prior to data collection, written informed consent was obtained from all participants who agreed to participate in the study and for the minors under the age of 18 years their legal representatives signed an informed consent form. They were informed of the study’s purpose and the voluntary nature of their participation, including the option to withdraw at any time. They were also assured that their data would remain confidential and be used only for research purposes. Once participants gave their consent, they completed the questionnaire in presence of research assistants who were on hand to answer any questions and address any immediate concerns.

## 2.4. Statistical analysis of data

Data analysis was performed using IBM SPSS 26 software. The normality of the data was assessed and considered normal based on skewness values between -2 and +2 and kurtosis values between -7 and +7 [41,42]. Validity of the study instrument was scrutinized and improved through expert judgment by the researchers [43]. On the other hand, questionnaire reliability was evaluated using Cronbach’s alpha. Acceptable internal reliability was established in advance for each measure used at.70 [44]. Descriptive statistics such as mean and standard deviation were calculated, and bivariate correlations were examined using Spearman’s correlation to explore the relationships between variables. Mann-Whitney U tests were used to evaluate significant differences in doping attitudes based on age and gender. To assess significant

differences in doping attitudes across three levels of running experience, the Kruskal-Wallis H test was applied. A significance level of p < 0.05 was adopted.

## 3. Results

### 3.1. Descriptive statistics

The participants were categorized by age: junior runners (14–20 years) and senior runners (20–45 years). Running length of experience was classified as short (1–7 years), medium (8–14 years), and long (15–21 years). Descriptive statistics and demographic differences in doping attitudes were analyzed using the Mann-Whitney U test and Kruskal-Wallis test, as shown in Table 1.

The results of the Mann-Whitney U test indicated no significant differences by age ($U$ = 11582.500, $p$ <.191) and gender ($U$ = 11437.500, $p$ <.827) on athletes' attitudes towards doping. Similarly, Kruskal- Wallis H test showed no significant differences by athlete's length of experience on attitudes towards doping across the different levels of running experience ($\chi2$ (2) = 1.359, $p$ <.507).

Table 2 presents the mean values, standard deviations, and normal distribution of the study variables, based on the values of the skewness and kurtosis coefficients. Additionally, the internal reliability coefficients for all subscales of the questionnaire are provided.

Results returned a mean and standard deviation of (4.14±.65) in task orientation, ego orientation (3.07±.79), mastery climate (4.17±.62), performance climate (2.88±.62) and (2.32±.70) in attitudes towards doping. The results of Skewness and kurtosis tests show that the distribution of data fall below -2 to +2, and the kurtosis values fall below -7 to +7. This indicate that the data was not normally distributed and this necessitated the choice of non-parametric inferential tests [41,42]. The questionnaires was considered reliable as the Cronbach's alpha values for the most variables were above the acceptable level of 0.70 [44,45].

**Table 1. Descriptive statistics and Demographic Differences in Doping Attitudes (N=323).**

| Mann Whitney U Test | | | | | | |
|---|---|---|---|---|---|---|
| Variables | Class | N | Mean rank | Sum of Ranks | U | P |
| Age | Junior runners | 134 | 153.94 | 20627.50 | 11582.500 | .191 |
| | Senior runners | 189 | 167.72 | 31698.50 | | |
| Gender | Male | 215 | 161.20 | 34657.50 | 11437.500 | .827 |
| | Female | 108 | 163.60 | 17668.50 | | |
| Kruskal-Wallis Test | | | | | | |
| Variable | Class | N | Mean rank | χ2 | Df | P |
| Running experience | Short Length | 243 | 159.61 | 1.359 | 2 | .507 |
| | Medium Length | 65 | 173.50 | | | |
| | Long Length | 15 | 150.90 | | | |

**Table 2. Descriptive statistics and internal reliability coefficients.**

| Variable | Mean ± SD | Skewness | Kurtosis | Cronbach α |
|---|---|---|---|---|
| Task Orientation | (4.14±.65) | -.921 | 1.170 | 0.66 |
| Ego Orientation | (3.07±.79), | -.118 | .237 | 0.47 |
| Mastery Climate | (4.17±.62), | -1.338 | 2.784 | 0.82 |
| Performance Climate | (2.88±.62) | -.059 | 086 | 0.76 |
| Doping Attitudes | (2.32±.70) | .518 | 0.612 | 0.89 |

Following the presentation of descriptive statistics, the subsequent section examines the relationships between the demographic variables and attitudes towards doping utilizing correlation analysis.

### 3.2. Correlations among study variables

The results in Table 3 indicate that there was a significant inverse relationship between task orientation and doping attitude ($rho$ = -.158; $p$ =.004) and a significant positive correlation between ego orientation and doping attitude ($rho$ =.362; $p$ <.001) in goal orientation. In motivational climate the results indicated significant inverse relationship between mastery climate and doping attitude ($rho$ = -.242; $p$ <.001) and a significant positive correlation between performance climate and doping attitude ($rho$ =.362; $p$ <.001).

## 4. Discussion

The purpose of this study was to examine the relationships between dispositional goal orientation and coach-created motivational climate on attitudes towards doping among Kenyan endurance runners. The study also established the differences of selected demographic variables of age, gender, and length of running experience on attitudes towards doping. Kenyan endurance runners have been reported to have been associated with the use of doping or performance-enhancing substances (PES) in sports. The study hypothesized that a task goal orientation and a mastery climate would be associated with negative attitudes towards doping, while an ego goal orientation and a performance climate would be associated with positive attitudes towards doping.

The findings of this study revealed that Kenyan endurance runners demonstrated negative attitudes towards doping, with low scores on attitudes towards doping. This suggests a low inclination towards engaging in doping behaviors and practices among these athletes, aligning with previous studies associating negative doping attitudes with anti-doping stances in sports [8,10,26]. Further, the study found no significant differences in attitudes towards doping based on gender, age, or length of athletic experience among Kenyan endurance runners. The lack of significant differences among Kenyan endurance runners could be due to the similar coaching environments and opportunities for competition, education and training that both male and female athletes receive irrespective of age and athletes' years of running experience. This finding aligns with previous research in Africa where gender did not significantly influence athletes' attitudes towards doping [24–26]. However, it contrasts with research outside Africa, which has often shown male athletes to have more permissive attitudes towards doping than female athletes [21–23], underscoring the unique cultural and contextual dynamics within Kenyan athletics that merit further investigation.

**Table 3. Correlations between the study variables (N=323).**

| | Variable | | 1 | 2 | 3 | 4 | 5 |
|---|---|---|---|---|---|---|---|
| 1 | Task Orientation | *(rho)* | 1.000 | .219** | .684** | -.103 | -.158** |
| | | *Sig.* (2-tailed) | . | .000 | .000 | .064 | .004 |
| 2 | Ego Orientation | *(rho)* | .219** | 1.000 | .092 | .396** | .362** |
| | | *Sig.* (2-tailed) | .000 | . | .099 | .000 | .000 |
| 3 | Mastery Climate | *(rho)* | .684** | .092 | 1.000 | -.076 | -.242** |
| | | *Sig.* (2-tailed) | .000 | .099 | . | .171 | .000 |
| 4 | Performance Climate | *(rho)* | -.103 | .396** | -.076 | 1.000 | .362** |
| | | *Sig.* (2-tailed) | .064 | .000 | .171 | . | .000 |
| 5 | Doping Attitudes | *(rho)* | -.158** | .362** | -.242** | .362** | 1.000 |
| | | *Sig.* (2-tailed) | .004 | .000 | .000 | .000 | . |
| | **. Correlation is significant at the 0.01 level (2-tailed). | | | | | | |

Consistent with previous studies [10,26], this study found that Kenyan endurance runners exhibited a high task orientation, compared to a lower ego orientation, suggesting that Kenyan endurance runners value effort, personal improvement, and learning over comparing their performance to others in judging their athletic competence and success. Furthermore, the significant inverse relationship between task orientation and doping attitudes, alongside a significant positive correlation between ego orientation and doping attitudes, aligns with previous research which has associated task-oriented athletes to negative doping attitudes and linking ego orientation to positive doping intentions and risky behaviors in sports [10,12,13].This finding highlights the importance of fostering a task-oriented mindset among the athletes, as such orientations emphasize personal improvement, effort and ethical competition devoid of doping over winning at all costs.

Additionally, the findings of this study revealed that Kenyan endurance runners perceived their coach motivational climate as being more mastery-focused compared to performance-focused, indicating that Kenyan coaches emphasize cooperative learning, individual development, and effort, rather than intra-team competition, punishment of mistakes or favoritism and external validation. Moreover, the findings of this study revealed that mastery climate was negatively associated with doping attitudes, while a performance-focused climate was positively correlated with doping attitudes, aligning with previous studies that have associated mastery climate with positive behavioral outcomes, including adherence to anti-doping norms [10,16,17]. These finding among Kenyan endurance runners underscore the significance of coaching environments and anti-doping education in shaping athletes' attitudes towards doping.

## 4.1. Limitations and Future Research

The study offers valuable insights into the relationships between dispositional goal orientation, motivational climate, and attitudes towards doping among Kenyan endurance runners. However, there are several limitations that warrants further research. First, the study was confined to Elgeyo-Marakwet County, limiting its generalizability to Kenyan endurance runners from other prominent counties. Future research should replicate the current study across other prominent Kenyan counties to provide a more comprehensive understanding of attitudes towards doping and inform broader policy and actions. Secondly, future studies could delve into examining the impact of ongoing education and preventive measures on attitudes towards doping in order to inform targeted interventions to promote clean sports. Thirdly, longitudinal studies could offer insights into the long-term effects of motivational climate and goal orientation on athletes' attitudes towards doping providing a more nuanced understanding of how these factors evolve over time and their implications for anti-doping efforts. Lastly, researchers could explore the perspectives of other stakeholders involved in performance sports, such as coaches, governing bodies, and support staff in understanding their perceptions and roles in promoting anti-doping attitudes in order to have a more comprehensive interventions aimed at fostering a culture of clean and fair competition in sports. Therefore, by addressing these limitations and pursuing future research avenues, scholars can contribute to a more comprehensive understanding of doping attitudes and develop targeted interventions to promote clean and fair competition in sports.

## 5. Conclusions

The study concludes that task orientation and mastery climate were found to be negatively associated with doping attitudes, while ego orientation and performance climate were positively associated with doping attitudes among Kenyan endurance runners. These findings hold important implications for coaches, sports organizations, and policymakers in athletics, suggesting the need for anti-doping interventions and strategies that promote task orientation and mastery climates. Such initiatives can help cultivate attitudes against doping, promote ethical sportsmanship, and reduce the risk of doping among athletes. Additionally, the study underscores the importance of implementing and maintaining anti-doping deterrence measures, including ongoing education and prevention efforts from organizations like the Anti-Doping Agency of Kenya (ADAK) and Athletics Kenya (AK) across all gender, age groups and levels of running experience to enhance clean sport among Kenyan endurance runners.

## Author contributions

**Conceptualization:** Kevin Kiprotich Kipchumba.

**Data curation:** Kevin Kiprotich Kipchumba.

**Formal analysis:** Kevin Kiprotich Kipchumba, Elijah Rintaugu, Francis Mwangi.

**Methodology:** Kevin Kiprotich Kipchumba, Elijah Rintaugu, Francis Mwangi, Benson Gathoni.

**Software:** Benson Gathoni.

**Supervision:** Elijah Rintaugu, Francis Mwangi.

**Writing – original draft:** Kevin Kiprotich Kipchumba.

**Writing – review & editing:** Elijah Rintaugu, Francis Mwangi, Benson Gathoni.

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
