## [Decision Letter · Decision Letter 0]

12 Jul 2024

PONE-D-24-23784Dispositional Goal Orientation and Perceptions of Coach Motivational Climate on Attitudes towards Doping among Kenyan Endurance RunnersPLOS ONE

Dear Dr. Kipchumba,

Thank you for submitting your manuscript to PLOS ONE. After careful consideration, we feel that it has merit but does not fully meet PLOS ONE’s publication criteria as it currently stands. Therefore, we invite you to submit a revised version of the manuscript that addresses the points raised during the review process.

**ACADEMIC EDITOR: **

**Overall Evaluation:**

The manuscript is exceptionally composed, tackling a noteworthy issue in the realm of sports, and offers helpful knowledge of the associations among objective orientations, motivational climate, and doping propensities among Kenyan endurance runners. With**minor** adjustments and clarifications, it has the potential to make a significant commitment to the writing on doping counteraction.

**Summary of Recommendations for Authors**

Enhance the clarity of the**research gap** and organize the literature review more effectively.Explain the reliability and validity of the instruments utilized. Additionally, discuss the justification for selecting non-parametric tests.Present statistical values and discuss their practical significance.Extend the practical implications and provide specific recommendations.Ensure proper formatting of references.To improve the overall quality of your writing, it may be necessary to re-evaluate the English language standards employed in your manuscript.

We look forward to receiving your revised manuscript.

Kind regards,

Clementswami Sukumaran, PhD

Academic Editor

PLOS ONE

2. In this instance it seems there may be acceptable restrictions in place that prevent the public sharing of your minimal data. However, in line with our goal of ensuring long-term data availability to all interested researchers, PLOS’ Data Policy states that authors cannot be the sole named individuals responsible for ensuring data access (http://journals.plos.org/plosone/s/data-availability#loc-acceptable-data-sharing-methods).

3. We note that you have referenced

(Chebet, S. (2014). Evaluation of Knowledge, Attitudes, and Practices of Doping among Elite Middle and Long Distance Runners in Kenya [Unpublished Ph.D. Thesis]. Kenyatta University.

Kamenju, J. W. (2011). Influence of Sports Disciplines and Demographics of Kenya’s Colleges Athletes On Their Awareness, Perception And Attitude To Performance-Enhancing Substances Use [Unpublished Ph.D Thesis]. Kenyatta University.)

which has currently not yet been accepted for publication. Please remove this from your References and amend this to state in the body of your manuscript: (ie “Bewick et al. [Unpublished]”) as detailed online in our guide for authors

Reviewers' comments:

Reviewer's Responses to Questions

**Comments to the Author**

1. Is the manuscript technically sound, and do the data support the conclusions?

Reviewer #1: Yes

2. Has the statistical analysis been performed appropriately and rigorously? 

Reviewer #1: Yes

3. Have the authors made all data underlying the findings in their manuscript fully available?

Reviewer #1: Yes

4. Is the manuscript presented in an intelligible fashion and written in standard English?

Reviewer #1: No

5. Review Comments to the Author

Reviewer #1: I am thankful for providing me an opportunity to review the manuscript. I believe the authors tap into an important topic in sports sciences, especially coach motivational climate on attitudes towards doping for athletes. However, I think some things need clarifying for the publication that will help in the interpretation and understanding of the general and specific concepts.

Introduction: In introduction first line remove the repetition words “Doping is a is a worldwide” Authors need to recheck the introduction. The language should be reviewed by a native speaker to improve clarity.

Method: Kindly justify which method used for the sample size/ and the procedure.

Discussion:

- Some explanations are repetitive with the results section.

- If the authors can discuss the mechanisms underlying the significant and non-significant results in more depth, it would be beneficial. Simply mentioning consistent and inconsistent results with previous investigations may not be engaging enough for readers.

Conclusion:

- When discussing the underlying mechanism, the section could be better organized.

6. PLOS authors have the option to publish the peer review history of their article (what does this mean? ). If published, this will include your full peer review and any attached files.

**Do you want your identity to be public for this peer review?** For information about this choice, including consent withdrawal, please see our Privacy Policy .

Reviewer #1: **Yes: ** MARRIUM BASHIR

---

## [Author Response · Author response to Decision Letter 1]

3 Sep 2024

Kevin K. Kipchumba,

Kenyatta University,

Physical Education and Exercise Science Department,

P. O Box 43844-00100,

Nairobi, Kenya.

Mobile: +254 723 716 230 or +254 708496708

Email: kevinkipchumba@gmail.com or Kipchumba.kevin@ku.ac.ke

August 19, 2024

Editor-in-Chief

PLOS

1265 Battery Street, Suite 200

San Francisco, CA 94111

United States

Dear Editor-in-Chief,

RE: RESPONSE TO REVIEWERS

I am writing to respond to reviewers and academic editor comments in regards to Research Manuscript ID PONE-D-24-23784 entitled “Dispositional Goal Orientation and Perceptions of Coach Motivational Climate on Attitudes towards Doping among Kenyan Endurance Runners”. We appreciate the insightful comments and suggestions provided by the academic editor and reviewers. We have carefully considered each point raised and have made the necessary revisions to enhance the quality and clarity of our manuscript. Below, we outline our responses to the specific concerns and describe how each has been addressed.

Response to Academic Editor's Comments

1. Enhance the clarity of the research gap and organize the literature review more effectively

We have revised the Introduction and Literature Review sections to clearly articulate the research gap and ensure a more logical organization of the literature. These changes have enhanced the focus and clarity of the manuscript, making the research gap more explicit.

2. Explain the reliability and validity of the instruments utilized. Additionally, discuss the justification for selecting non-parametric tests.

The reliability and validity of the instruments have been addressed by providing a detailed discussion in the Methodology section. Specifically, the reliability of the questionnaires has been supported by Cronbach’s alpha values, which were above the acceptable level of 0.70 for most variables (citing Nunnally, 1978; Shaughnessy et al., 2003).

In order to justify the selection of non-parametric tests, we have cited the results of skewness and kurtosis tests, which indicated that the data was not normally distributed. The skewness values fall below -2 and +2, and kurtosis values fell below -7 and +7, which necessitated the use of non-parametric inferential tests (Hair et al., 2010; Bryne, 2010). These revisions are reflected in the Methods section of our manuscript.

3. Present statistical values and discuss their practical significance.

In response to the reviewers' comments, the Results of the study were previously addressed in the Discussion sections. After reading relevant literature and PLOS ONE guidelines, we believe that our manuscript has included the relevant statistical values and more in-depth discussion of their practical significance.

4. Extend the practical implications and provide specific recommendations.

The Conclusion section contain a more detailed practical implications and specific recommendations based on our findings offering more actionable insights for researchers, coaches, and sports organizations

5. Ensure proper formatting of references.

We have reviewed and corrected the reference list to ensure it meets PLOS ONE's formatting requirements. Unpublished Ph.D. theses have been removed, and all references have been thoroughly checked for accuracy and completeness.

6. To improve the overall quality of your writing, it may be necessary to re-evaluate the English language standards employed in your manuscript.

We have revised the manuscript to improve the clarity and quality of the English language. This involved rephrasing sentences for better readability and correcting any grammatical errors.

Specific Response to Reviewer's Comments:

1. Introduction

We have revised the first line of the introduction to eliminate the repetition and have reviewed the entire section to ensure clarity and coherence. The language has been polished for improved readability.

2. Method

We have justified the method used for determining the sample size and provided a more detailed explanation of the procedures followed. This information has been added to the Methods section for greater transparency.

3. Discussion

We have removed repetitive content between the Discussion and Results sections and expanded the mechanisms underlying both significant and non-significant results to provide a deeper understanding of the findings.

Additional Journal-Specific Requirements:

1. Data Availability:

In compliance with the journal’s data policy, we have provided contact information for a non-author institutional body that can respond to data access requests. This information is included in the Contact information.

We have uploaded a revised manuscript with tracked changes, an unmarked clean copy, and this rebuttal letter as separate files. We trust that these revisions address all the concerns raised and hope that the revised manuscript is now suitable for publication in PLOS ONE.

Thank you once again for your constructive feedback and for considering our manuscript for publication.

Yours sincerely,

Kevin Kipchumba

Kenyatta University, Nairobi, Kenya

---

## [Decision Letter · Decision Letter 1]

22 Sep 2024

PONE-D-24-23784R1Dispositional Goal Orientation and Perceptions of Coach Motivational Climate on Attitudes towards Doping among Kenyan Endurance RunnersPLOS ONE

Dear Dr. Kipchumba,

Thank you for submitting your manuscript to PLOS ONE. After careful consideration, we feel that it has merit but does not fully meet PLOS ONE’s publication criteria as it currently stands. Therefore, we invite you to submit a revised version of the manuscript that addresses the points raised during the review process.

**ACADEMIC EDITOR:** Your paper, "Dispositional Goal Orientation and Perceptions of Coach Motivational Climate on Attitudes towards Doping among Kenyan Endurance Runners," is well-structured, but several areas could be slightly improved:

**Flow of Information** : The transition from the literature review to the methodology is abrupt. A concluding paragraph in the literature review would create a smoother transition.**Coherence** : The Results section would benefit from linking sentences between descriptive statistics and correlation analysis for better readability.**Clarity** : The cross-sectional survey design and sampling method need more detail. Clarifying the stratified random sampling process will improve precision.**Structure** : Consolidating demographic findings into a single subsection would enhance focus.**Logical Gaps** : Before offering recommendations, discuss the findings’ implications in the context of existing literature for a stronger conclusion.

These revisions will enhance the paper's clarity, coherence, and impact. 

We look forward to receiving your revised manuscript.

Kind regards,

Clementswami Sukumaran, PhD

Academic Editor

PLOS ONE

Journal Requirements:

Reviewers' comments:

Reviewer's Responses to Questions

**Comments to the Author**

1. If the authors have adequately addressed your comments raised in a previous round of review and you feel that this manuscript is now acceptable for publication, you may indicate that here to bypass the “Comments to the Author” section, enter your conflict of interest statement in the “Confidential to Editor” section, and submit your "Accept" recommendation.

Reviewer #1: All comments have been addressed

Reviewer #2: All comments have been addressed

2. Is the manuscript technically sound, and do the data support the conclusions?

Reviewer #1: Yes

Reviewer #2: Yes

3. Has the statistical analysis been performed appropriately and rigorously? 

Reviewer #1: Yes

Reviewer #2: Yes

4. Have the authors made all data underlying the findings in their manuscript fully available?

Reviewer #1: Yes

Reviewer #2: Yes

5. Is the manuscript presented in an intelligible fashion and written in standard English?

Reviewer #1: Yes

Reviewer #2: Yes

6. Review Comments to the Author

Reviewer #1: (No Response)

Reviewer #2: The paper "Dispositional Goal Orientation and Perceptions of Coach Motivational Climate on Attitudes towards Doping among Kenyan Endurance Runners" examines the influence of goal orientation and coach-created motivational climate on doping attitudes among Kenyan endurance runners. While the paper demonstrates effective organization, several areas warrant improvement:

1. Flow of Information: The transition between the literature review and methodology sections lacks cohesion. The inclusion of a summarizing paragraph at the conclusion of the literature review would facilitate a smoother transition.

2. Coherence Between Paragraphs: The Results section would benefit from enhanced readability through the incorporation of linking sentences between descriptive statistics and correlation analysis.

3. Clarity: The description of the cross-sectional survey and sampling method lacks precision. Elucidating the utilization of stratified random sampling would enhance methodological clarity.

4. Structure and Organization: Demographic findings pertaining to doping attitudes are dispersed throughout the paper. Consolidating these findings under a single subsection would improve focus and coherence.

5. Logical Gaps: The conclusion section progresses directly to recommendations without adequate discussion of implications. Incorporating a discussion of findings within the context of existing literature prior to presenting recommendations would provide more comprehensive insights.

Addressing these aspects would enhance the paper's clarity, coherence, and overall impact.

7. PLOS authors have the option to publish the peer review history of their article (what does this mean? ). If published, this will include your full peer review and any attached files.

**Do you want your identity to be public for this peer review?** For information about this choice, including consent withdrawal, please see our Privacy Policy .

Reviewer #1: **Yes: ** MARRIUM BASHIR

Reviewer #2: **Yes: ** Clementswami Sukumaran, PhD

---

## [Author Response · Author response to Decision Letter 2]

30 Nov 2024

Kevin K. Kipchumba,

Kenyatta University,

Physical Education and Exercise Science Department,

P. O Box 43844-00100,

Nairobi, Kenya.

Mobile: +254 723 716 230 or +254 708496708

Email: kevinkipchumba@gmail.com or Kipchumba.kevin@ku.ac.ke

November 29, 2024

Editor-in-Chief

PLOS

1265 Battery Street, Suite 200

San Francisco, CA 94111

United States

Dear Editor-in-Chief,

RE: RESPONSE TO REVIEWERS

I am writing to respond to reviewers and academic editor comments in regards to Research Manuscript ID PONE-D-24-23784 entitled “Dispositional Goal Orientation and Perceptions of Coach Motivational Climate on Attitudes towards Doping among Kenyan Endurance Runners”. We appreciate the insightful comments and suggestions provided by the academic editor and reviewers. We have carefully considered each point raised and have made the necessary revisions to enhance the quality and clarity of our manuscript. Below, we outline our responses to the specific concerns and describe how each has been addressed.

Response to Academic Editor's Comments

1. Flow of Information: The transition from the literature review to the methodology is abrupt. A concluding paragraph in the literature review would create a smoother transition

We have revised the Literature Review and methodology sections with concluding paragraph at the end of the literature review section to create smoother transition to methodology section.

2. Coherence: The Results section would benefit from linking sentences between descriptive statistics and correlation analysis for better readability.

These has been addressed by linking the two areas in the results section. These revisions are reflected in the Results section of our manuscript.

3. Clarity: The cross-sectional survey design and sampling method need more detail. Clarifying the stratified random sampling process will improve precision.

These concern from the reviewers have been addressed by rephrasing the sentence and adding additional information to read “The population was divided into meaningful strata based on key characteristics: gender, age group (junior and senior runners), and length of running experience. Within each stratum, participants were then randomly selected to account for variations in these demographic factors. Stratified random sampling method allowed for balanced representation across subgroups, enhancing the precision and generalizability of findings to the broader population of Kenyan endurance runners”. We believe this will improve on the precision.

4. Structure: Consolidating demographic findings into a single subsection would enhance focus.

These has been addressed by consolidating the two areas in the results section. These revisions are reflected in the Results section of our manuscript.

5. Logical Gaps: Before offering recommendations, discuss the findings’ implications in the context of existing literature for a stronger conclusion.

We have revised our manuscript to address logical gaps by discussing the findings’ implications within the context of existing literature, thereby strengthening the conclusion and recommendations.

Thank you once again for your constructive feedback and for considering our manuscript for publication.

Yours sincerely,

Kevin Kipchumba

Kenyatta University, Nairobi, Kenya

---

## [Editor Report · Decision Letter 2]

3 Dec 2024

PONE-D-24-23784R2Dispositional Goal Orientation and Perceptions of Coach Motivational Climate on Attitudes towards Doping among Kenyan Endurance RunnersPLOS ONE

Dear Dr. Kipchumba,

Thank you for submitting your manuscript to PLOS ONE. After careful consideration, we feel that it has merit but does not fully meet PLOS ONE’s publication criteria as it currently stands. Therefore, we invite you to submit a revised version of the manuscript that addresses the points raised during the review process. ==============================

**ACADEMIC EDITOR: **

**Dear Authors,**

Thank you for your submission and for addressing the comments from the previous review round. The manuscript is well-organized, and the data support the conclusions drawn. However, there are a few areas that require further attention to improve the clarity and flow of the manuscript:

Completed Comments:

1. Is the manuscript technically sound, and do the data support the conclusions? 

    Completed: The reviewer has confirmed that the manuscript presents a technically sound piece of research and that the data support the conclusions.

2. Has the statistical analysis been performed appropriately and rigorously? 

    Completed: The reviewer confirms that the statistical analysis is appropriately performed.

3. Have the authors made all data underlying the findings in their manuscript fully available? 

    Completed: The reviewer confirms that the data availability statement complies with PLOS ONE's data policy.

4. Is the manuscript presented in an intelligible fashion and written in standard English? 

    Completed: The reviewer indicates that the manuscript is written clearly in standard English, with no notable typographical or grammatical errors.

5. If the authors have adequately addressed your comments raised in a previous round of review and you feel that this manuscript is now acceptable for publication, you may indicate that here. 

    Completed: The reviewer acknowledges that all comments from the previous review have been addressed.

** Incomplete Comments (Areas for Improvement):**

 1. Flow of Information (Incomplete) 

Reviewer's Comment: 

The transition between the literature review and methodology sections lacks cohesion. The inclusion of a summarizing paragraph at the conclusion of the literature review would facilitate a smoother transition.

 Required Action: 

  Add a concluding summarizing paragraph at the end of the literature review to bridge the gap to the methodology section. This paragraph should briefly restate the literature's main findings and gaps, followed by a segue into your research focus.

 Example to include: 

  “While existing research has explored several factors influencing doping behaviors in athletes, there is a clear gap in understanding how dispositional goal orientation and coach created motivational climate influence doping attitudes, particularly in Kenyan endurance runners. This study aims to fill this gap by examining these factors within the Kenyan athletic context.”

2. Coherence Between Paragraphs (Undone) 

Reviewer's Comment: 

The Results section would benefit from enhanced readability through the incorporation of linking sentences between descriptive statistics and correlation analysis.

What needs to be done: 

Add transition sentences between sections in the Results chapter to enhance readability and link different analysis parts.

Example to include: 

After the descriptive statistics section, include a statement such as: 

"Following the presentation of descriptive statistics, the subsequent section examines the relationships between the demographic variables and attitudes towards doping utilizing correlation analysis."

This will facilitate the reader's transition from one analysis to the next.

3. Clarity (Undone) 

Reviewer's Comment: 

The description of the cross-sectional survey and sampling method lacks precision. Elucidating the utilization of stratified random sampling would enhance methodological clarity.

What needs to be done: 

Provide additional detail regarding the description of stratified random sampling in the Methodology section. Elucidate the process of participant selection within each stratum.

Example to include: 

"A stratified random sampling technique was employed to ensure proportional representation across gender, age, and experience. Initially, the population was stratified based on these demographic variables. Subsequently, participants were randomly selected from each stratum, ensuring the sample accurately reflected the target population of Kenyan endurance runners."

This clarifies the methodology for readers unfamiliar with the sampling technique.

4. Structure and Organization (Undone) 

Reviewer's Comment: 

Demographic findings pertaining to doping attitudes are dispersed throughout the paper. Consolidating these findings under a single subsection would improve focus and coherence.

What needs to be done: 

Establish a dedicated subsection for demographic differences in doping attitudes. This will facilitate the consolidation of all related findings in one location, enhancing readability.

Example of a new subsection: 

Add a section titled "Demographic Differences in Doping Attitudes" under Results and group all related findings (Mann-Whitney U test and Kruskal–Wallis test results) together.

5. Logical Gaps (Undone) 

Reviewer's Comment: 

The conclusion section progresses directly to recommendations without adequate discussion of implications. Incorporating a discussion of findings within the context of existing literature prior to presenting recommendations would provide more comprehensive insights.

What needs to be done: 

Expand the discussion section by linking your findings with existing literature before offering recommendations. This will provide a more thorough analysis and improve the flow of the conclusion.

Example to include: 

"Consistent with previous studies (Allen et al., 2015; Mwangi et al., 2019), this research found that task orientation was negatively associated with attitudes towards doping, suggesting that an emphasis on personal growth and effort can discourage doping behaviors. However, the contrasting findings regarding gender and age highlight the need for more contextual research in the Kenyan athletic setting…"

This approach connects your findings to the existing body of knowledge before proceeding to any practical recommendations.

**Summary:**

Done:

 The manuscript is technically sound, with appropriate statistical analysis and data availability.

 Language and presentation are clear and correct.

 The review comments from the previous round have been addressed.

Undone:

1. Flow of Information: Add a summarizing paragraph to the literature review to transition smoothly to the methodology.

2. Coherence Between Paragraphs: Include linking sentences between descriptive statistics and correlation analysis in the Results section.

3. Clarity: Provide more precision in describing the stratified random sampling method.

4. Structure and Organization: Consolidate demographic findings regarding doping attitudes under one subsection.

5. Logical Gaps: Expand the conclusion section by discussing findings in the context of existing literature before offering recommendations.

Once these minor revisions are implemented, I am confident the manuscript will be suitable for publication. I look forward to receiving your updated submission.

We look forward to receiving your revised manuscript.

Kind regards,

Clementswami Sukumaran, PhD

Academic Editor

PLOS ONE
---

## [Author Response · Author response to Decision Letter 3]

17 Feb 2025

Kevin K. Kipchumba,

Kenyatta University,

Physical Education and Exercise Science Department,

P. O Box 43844-00100,

Nairobi, Kenya.

Mobile: +254 723 716 230 or +254 708496708

Email: kevinkipchumba@gmail.com or Kipchumba.kevin@ku.ac.ke

February 13, 2025

Editor-in-Chief

PLOS

1265 Battery Street, Suite 200

San Francisco, CA 94111

United States

Dear Editor-in-Chief,

RE: RESPONSE TO REVIEWERS

I am writing to respond to reviewers and academic editor comments in regards to Research Manuscript ID PONE-D-24-23784 entitled “Dispositional Goal Orientation and Perceptions of Coach Motivational Climate on Attitudes towards Doping among Kenyan Endurance Runners”. We appreciate the insightful comments and suggestions provided by the academic editor and reviewers. We have carefully considered each point raised and have made the necessary revisions to enhance the quality and clarity of our manuscript. Below, we outline our responses to the specific concerns and describe how each has been addressed.

Response to Academic Editor's Comments

1. Flow of Information: The transition from the literature review to the methodology is abrupt. A concluding paragraph in the literature review would create a smoother transition

We have revised the Literature Review and methodology sections with concluding paragraph at the end of the literature review section guided by reviewer statements to create smoother transition to methodology.

2. Coherence Between Paragraphs:

These has been addressed by linking the two areas in the results section guided by reviewer statements. These revisions are reflected in the Results section of our manuscript.

3. Clarity:

These concern from the reviewers have been addressed by rephrasing the sentence and adding guided information from the reviewers to improve on the clarity.

4. Structure:

These has been addressed by consolidating the two areas in the results section. These revisions are reflected in the Results section of our manuscript.

5. Logical Gaps:

We have revised our manuscript to address logical gaps by discussing the findings’ implications within the context of existing literature and adding guiding statements from the reviewers, thereby strengthening the conclusion and recommendations.

Thank you once again for your constructive feedback and for considering our manuscript for publication.

Yours sincerely,

Kevin Kipchumba

Kenyatta University, Nairobi, Kenya

---

## [Decision Letter · Decision Letter 3]

5 Mar 2025

Dispositional Goal Orientation and Perceptions of Coach Motivational Climate on Attitudes towards Doping among Kenyan Endurance Runners

PONE-D-24-23784R3

Dear Dr. Kipchumba,

We’re pleased to inform you that your manuscript has been judged scientifically suitable for publication and will be formally accepted for publication once it meets all outstanding technical requirements.

Kind regards,

Clementswami Sukumaran, PhD

Academic Editor

PLOS ONE

Additional Editor Comments (optional):

Reviewers' comments:

Reviewer's Responses to Questions

**Comments to the Author**

1. If the authors have adequately addressed your comments raised in a previous round of review and you feel that this manuscript is now acceptable for publication, you may indicate that here to bypass the “Comments to the Author” section, enter your conflict of interest statement in the “Confidential to Editor” section, and submit your "Accept" recommendation.

Reviewer #2: All comments have been addressed

2. Is the manuscript technically sound, and do the data support the conclusions?

Reviewer #2: Yes

3. Has the statistical analysis been performed appropriately and rigorously? 

Reviewer #2: Yes

4. Have the authors made all data underlying the findings in their manuscript fully available?

Reviewer #2: Yes

5. Is the manuscript presented in an intelligible fashion and written in standard English?

Reviewer #2: Yes

6. Review Comments to the Author

Reviewer #2: Dear Authors,

This correspondence serves to confirm receipt of your revised manuscript entitled "Dispositional Goal Orientation and Perceptions of Coach Motivational Climate on Attitudes towards Doping among Kenyan Endurance Runners." The following provides a summary of my comments and the current status of each:

1. Flow of Information – The transition between the literature review and methodology has been enhanced with the addition of a concluding paragraph. (Addressed)

2. Coherence in Results – Linking sentences have been incorporated to improve readability between descriptive statistics and correlation analysis. (Addressed)

3. Clarity in Methodology – The explanation of stratified random sampling has been elucidated. (Addressed)

4. Structure & Organization – Demographic findings on doping attitudes have been consolidated under a single subsection. (Addressed)

5. Logical Gaps in Conclusion – A discussion of findings within the context of existing literature has been included prior to presenting recommendations. (Addressed)

With all concerns adequately addressed, the manuscript now demonstrates improved cohesion, structure, and clarity.

It is deemed suitable for acceptance.

7. PLOS authors have the option to publish the peer review history of their article (what does this mean? ). If published, this will include your full peer review and any attached files.

**Do you want your identity to be public for this peer review?** For information about this choice, including consent withdrawal, please see our Privacy Policy .

Reviewer #2: No

---

## [Editor Report · Acceptance letter]

PONE-D-24-23784R3

PLOS ONE

Dear Dr. Kipchumba,

I'm pleased to inform you that your manuscript has been deemed suitable for publication in PLOS ONE. Congratulations! Your manuscript is now being handed over to our production team.

Kind regards,

on behalf of

Dr. Clementswami Sukumaran

Academic Editor

PLOS ONE